# Adsorption Characteristics of Phosphate Based on Al-Doped Waste Ceramsite: Batch and Column Experiments

**DOI:** 10.3390/ijerph20010671

**Published:** 2022-12-30

**Authors:** Yameng Ma, Jia Zhu, Jianghua Yu, Yicheng Fu, Chao Gong, Xiao Huang

**Affiliations:** 1Collaborative Innovation Center of Atmospheric Environment and Equipment Technology, Jiangsu Key Laboratory of Atmospheric Environment Monitoring and Pollution Control, School of Environmental Science and Engineering, Nanjing University of Information Science & Technology, Nanjing 210044, China; 2School of Materials and Environmental Engineering, Shenzhen Polytechnic, Shenzhen 518055, China; 3State Key Laboratory of Simulation and Regulation of River Basin Water Cycle, China Institute of Water Resources and Hydropower Research, Beijing 100038, China

**Keywords:** phosphate, ceramsite, adsorption, desorption, column experiment

## Abstract

Phosphorus widely existing in rainfall and wastewater impacts the water environment. In this study, sludge, cement block, and coal fly ash were employed as ceramsite material to synthesize Al-doped waste ceramsite (Al-ceramsite) for removing phosphate (PO_4_^3−^-P) from aqueous solutions. Batch static adsorption–desorption experiments were designed to investigate the effect of various parameters such as Al-ceramsite dosage, PO_4_^3−^-P concentration, temperature, initial pH, coexisting ions, and desorbents on the removal of PO_4_^3−^-P. Also, the fate of PO_4_^3−^-P removal efficiency in actual rainwater was studied through dynamic adsorption column experiments using Al-ceramsite. Results showed that Al-ceramsite could remove PO_4_^3−^-P efficiently under the optimum parameters as follows: Al-ceramsite dosage of 40 g/L, initial PO_4_^3−^-P concentration of 10 mg/L, temperature of 25 °C, and pH of 5. Besides that, the Al-ceramsite could completely remove PO_4_^3−^-P in actual rainwater, and the effluent PO_4_^3−^-P concentration was lower than the environmental quality standards for surface water Class Ⅰ (0.02 mg/L). The adsorption characteristics of Al-ceramsite on PO_4_^3−^-P by X-ray photoelectron spectroscopy (XPS) were further explained. As a result, ligand exchange and complexation were confirmed as the main PO_4_^3−^-P removal mechanism of Al-ceramsite. Thus, Al-ceramsite was prepared from industrial waste and has shown excellent potential for phosphorus removal in practical applications.

## 1. Introduction

Phosphorus, as an essential nutrient for biological growth, plays an irreplaceable role in modern agriculture and industrial production [1]. Driven by rainfall and wastewater discharges, excessive phosphorus runoff into nearby waters can lead to non-point source phosphorus pollution, such as eutrophication, algal blooms, reduction of biodiversity, deterioration of water quality, and human health problems [2]. Therefore, rigorous control of phosphorus load in waters has become a new challenge for water environment management.

Currently, the main methods such as biological methods, chemical precipitation, ion exchange, and adsorption have been widely applied to remove phosphorus from water [3]. The adsorption method is regarded as a good application prospect due to its simple, low-cost, and high efficiency of phosphorus removal [4]. In recent years, different types of adsorbents have been developed for phosphate (PO_4_^3−^-P) removal, including carbon-based materials [5], zeolites [6], silica [7], diatomaceous earth [8], and so on. However, the practical application was hindered by the limitations of the high cost and isolation difficulty. To achieve excellent PO_4_^3−^-P adsorption performance at an acceptable cost, industrial waste-based adsorbent materials have attracted wide attention [9]. A lot of work has been reported on the basis of industrial waste-based material to prepare adsorbents for the removal of phosphorus, including municipal sludge [10], coal fly ash [11], cement blocks [12], red mud [13], and slag [14]. Lin et al., (2021) reported the high adsorption capacity of red mud-based ceramsite for PO_4_^3−^-P [15]. A study by Gu et al., (2021) showed that coal fly ash achieved phosphorus removal through adsorption and precipitation [16]. Similarly, Liu et al., (2020) innovatively used waste concrete to enhance the removal of phosphorus, and demonstrated that the phosphorus adsorption capacity of modified waste concrete was up to 100 mg/g [17]. Industrial waste-based ceramsite can improve the level of resource utilization and address the issues of environmental pollution. This is a necessary condition for sustainable development strategy.

In China, more than 10 million tons of dry sludge was produced annually from wastewater treatment plants [9]. Sludge disposal by landfills and incinerators could not satisfy the trend of sustainable development in the future. The construction industry is reportedly the largest consumer of natural resources, producing an enormous amount of 2.65 billion tons of waste annually [18]. Moreover, finding ways of sustainable alternatives to reuse and recycle these industrial wastes is gaining importance due to the amount of wastes generated and concerns about inadequate final disposal. The porous structure and high specific surface area of coal fly ash and cement from industrial wastes make it possible to adsorb and precipitate phosphorus [19]. However, in order to further improve the adsorption capacity of adsorbents on pollutants, metal ions have been widely used in the surface modification of adsorbents [20]. Yang et al., (2018) pointed out that the iron-modification waste-activated sludge (WAS)-based biochar contributed to PO_4_^3−^-P adsorption, and the maximum PO_4_^3−^-P adsorption capacity of 111.0 mg/g was observed using FeCl_3_-impregnated WAS-based biochar [21]. Deng et al., (2021) used Mg-modified-biochar composites for removing phosphate from waste streams and reported the maximum phosphate adsorption capacity of 128.21 mg/g [22]. Using industrial wastes as raw material for the preparation, ceramsite is a sustainable development strategy which can not only improve industrial waste management, but also reduce environmental effects.

This study synthesized a composite ceramsite composed of sludge, cement blocks, and coal fly ash, which was further doped with aluminum salts to form Al-doped waste ceramsite (Al-ceramsite) to enhance PO_4_^3−^-P removal from aqueous solutions. The optimal process parameters were determined by a static adsorption experiment, and the desorption behavior of Al-ceramsite were further understood. The dynamic adsorption column was to gain a deeper understanding of the adsorption performance of Al-ceramsite in rainwater under more realistic operating conditions. Therefore, the adsorption mechanism of Al-ceramsite on PO_4_^3−^-P were analyzed by the X-ray photoelectron spectroscopy (XPS). With high efficiency and low cost, the waste-based ceramsite is of significant importance for phosphorus removal, and provides an economically sustainable way for rainwater purification.

## 2. Materials and Methods

### 2.1. Materials

All reagents were of analytical grade. Aluminum nitrate Al(NO_3_)_3_, sodium sulphate (Na_2_SO_4_), sodium chloride (NaCl), sodium phosphate (Na_3_PO_4_), ascorbic acid (C_6_H_8_O_6_), magnesium sulphate (MgSO_4_), and sodium fluoride (NaF) were bought from Aladdin Reagents Co., Ltd. (Shanghai, China). Potassium dihydrogen phosphate (KH_2_PO_4_), hydrochloric acid (HCl), and ammonium molybdate ((NH_4_)_2_MoO_4_) were purchased from Sinopharm Chemical Reagent Co., Ltd. (Shanghai, China). Sodium hydroxide (NaOH) and antimony potassium tartrate (C_4_H_4_KO_7_Sb·_0.5_H_2_O) were from Macklin Biochemical Co., Ltd. (Shanghai, China). Sodium acetate (NaAC) and sodium carbonate (Na_2_CO_3_) were from Bide Pharmatech Ltd. (Shanghai, China). Calcium oxide was from Meryer Chemical Technology Co., Ltd. (Shanghai, China). All solutions were prepared with deionized water. Sludge, cement block, and coal fly ash were used to prepare ceramsite. Sludge was collected from a sewage-treatment plant in Nanjing (Jiangsu, China), cement block was produced from a construction site in Nanjing (Jiangsu, China), and coal fly ash was acquired from a power plant in Nanjing (Jiangsu, China).

### 2.2. Preparation of Ceramsite

Al-ceramsite was prepared based on a previous study [23]. Preparation conditions were as following: firstly, sludge, cement blocks, and coal fly ash were crushed and ground by a predetermined mass ratio (3:2:6) and then fed into the granulator to form granules. Secondly, the particles of 4–6 mm were screened and dried at 105 °C for 2 h. Finally, undoped-ceramsite (un-ceramsite) was obtained by preheating at 350 °C for 10 min and calcining at 1150 °C. Al-ceramsite was prepared via doping Al(NO_3_)_3_ solution with ceramsite under the optimum conditions (Al(NO_3_)_3_ concentration of 0.75 mol/L, the ratio of ceramsite mass to Al(NO_3_)_3_ solution of 1:3, doping time of 3 h, and temperature of 25 °C). The Al-ceramsite was washed with distilled water to neutral and calcined at 600 °C for 4 h. After cooling, Al-ceramsite was collected and reserved. Of these, the physicochemical properties of Un-ceramsite and Al-ceramsite are shown in Table 1.

### 2.3. Experimental Setup

The static adsorption performance of ceramsite was tested in a beaker. A certain mass of ceramsite was added to the PO_4_^3−^-P solution and mixed in a constant temperature shaker (THZ-C, Taicang Qiangle Experimental Equipment Co., Ltd., Jiangsu, China) at 120 r/min. Dynamic adsorption experiments were tested in polyvinylchloride (PVC) tubes with an inner diameter of 3.5 cm and a height of 40 cm. Figure 1 reveals the setup of the static adsorption and the continuous flow dynamic adsorption column experiment. Experiments were performed at room temperature. Natural rainfall (rainwater was collected from natural rainfall in Nanjing, China, in the summer of July–August 2019, with a PO_4_^3−^-P content of 0.58 mg/L and a rainwater pH of 7.2–8.7) was selected as the influent water for the column experiment. Flow rate maintained by a peristaltic pump (BT100-2J, Longer Precision Pump Co., Ltd., Hebei, China). Samples were taken at scheduled times for determination of PO_4_^3−^-P concentration.

### 2.4. Experimental Procedure

#### 2.4.1. Static Adsorption of PO_4_^3−^-P by Al-Ceramsite

To investigate the effect of Al-ceramsite on PO_4_^3−^-P adsorption, Un-ceramsite and Al-ceramsite were added to 10 mg/L of PO_4_^3−^-P solutions (pH = 7 and temperature of 25 °C) for 20 h, respectively. The effect of Al-ceramsite on PO_4_^3−^-P removal was further studied in different environmental factors (dosage of Al-ceramsite, initial PO_4_^3^-P concentration, and initial pH). Effect of coexisting ions were evaluated by using different coexisting ions (Ca^2+^, Mg^2+^, SO_4_^2−^, CO_3_^2−^, F^−^, and Cl^−^), and the influence of varying coexisting ion concentrations (2–10 mmol/L) on PO_4_^3−^-P absorption was investigated.

#### 2.4.2. Effect of Phosphorus Desorption

The adsorption-saturated Al-ceramsite was obtained at about 25 °C, pH = 6.5, and 120 r/min for 48 h. Firstly, the desorption performance of Al-ceramsite was examined at pH = 4–10. Secondly, the effect of desorbent type (NaAC, Na_2_CO_3_, and NaOH) on the desorption of Al-ceramsite was investigated. Finally, the efficiency of desorbent concentration (0.25–2 mol/L) on the desorption of Al-ceramsite was analyzed.

#### 2.4.3. PO_4_^3−^-P Removal by Dynamic Adsorption Column

To understand the effect of Al-ceramsite on the adsorption of PO_4_^3−^-P from rainwater by column experiments, different submerged zone depths (SZD) and filter media heights (FMH) was investigated. Then, samples were collected at intervals for PO_4_^3−^-P concentration determination. Subsequently, the effect of stormwater biofilter on PO_4_^3−^-P removal in practical application was simulated, and the columns were filled with Al-ceramsite and gravel as fillers, respectively. The effect of columns on the removal of PO_4_^3−^-P from rainwater was analyzed in different influent flows (0.04–0.12 mL/s), and antecedent dry day (ADD) of 3–21 days, where SZD was 20 cm and FMH was 15 cm.

### 2.5. Analytical Methods

A pH meter (PHS-3C, Shanghai INESA Scientific Instrument Co, Ltd., Shanghai, China) was used to monitor the solution pH. The concentration of PO_4_^3−^-P was determined by ultraviolet spectrophotometer (GEN10S UV-VIS, Thermo Fisher Scientific Inc. Shanghai, China) and based on standard methods [24] as follows: PO_4_^3−^-P (molybdate colorimetric method). X-ray photoelectron spectroscopy (XPS) was obtained using an ESCALAB 250, Thermo instrument. Each batch was repeated three times using the mean value for analysis. Origin 2018 software was used to analyze the data.

The PO_4_^3−^-P adsorption capacity (Q_e_, mg/g) on ceramsite is calculated by Equation (1). The PO_4_^3−^-P removal rate (R, %) is calculated by Equation (2). The PO_4_^3−^-P desorption efficiency (D, %) is calculated by Equation (3).

Q_e_ = (C_0_ − C_e_) × V/m,
(1)


R (%) = (C_0_ − C_e_)/C_0_ × 100%,
(2)


D (%) = Q_ed_/Q_ts_ × 100%
(3)

where Q_e_ is the adsorption capacity (mg/g); V is the solution volume (L); C_0_ and C_e_ are the initial and equilibrium concentrations of PO_4_^3−^-P (mg/L), respectively; m is the mass of ceramsite (g); R is the PO_4_^3−^-P removal efficiency (%); Q_ed_ is the PO_4_^3−^-P desorption capacity; and Q_ts_ the theoretical saturation adsorption capacity.

## 3. Results and Discussion

### 3.1. Optimization of Adsorption Conditions

#### 3.1.1. Advantages of Al-Ceramsite

The adsorption of Al-ceramsite on PO_4_^3−^-P was shown in Figure 2a. It was evident that the adsorption performance of Al-ceramsite was much higher than that of Un-ceramsite at the same dosage. The rapid enhancement of the PO_4_^3−^-P adsorption process as the dosage of Al-ceramsite increases suggests that the removal efficiency of PO_4_^3−^-P was increased. The best removal efficiency result for PO_4_^3−^-P was achieved using 40 g/L of Al-ceramsite obtaining 83% of PO_4_^3−^-P removal, and the adsorption equilibrium was gradually reached. The results showed that the Al-ceramsite facilitated the improvement of the PO_4_^3−^-P adsorption capacity, which confirmed it has a broad application prospect.

The adsorbent materials for the phosphorus adsorption reported were compared in Table 2. The Al-ceramsite exhibited excellent adsorption ability for PO_4_^3−^-P. With regard to the adsorption capacity of Al-ceramsite, further improvement is needed compared to other adsorbent materials. However, the advantage of the materials derived from industrial waste is that Al-ceramsite has a relatively superior application potential for PO_4_^3−^-P treatment in rainwater. Thus, the surface roughness, porosity, and strength of ceramsite synthesized from sludge, cement blocks, and coal fly ash were improved [25,26]. After the ceramsite was doped with aluminum salt solution, PO_4_^3−^ and Al^3+^ generated stable phosphorus precipitation (Equation (4)), which further improved the PO_4_^3−^-P removal efficiency [27]. The fast adsorption speed of PO_4_^3−^-P in 40 g/L Al-ceramsite may be due to the increased solid–liquid contact area and provided more effective adsorption sites during the adsorption process [28,29]. However, with the increase of the dosage of Al-ceramsite, the surface adsorption sites became saturated and the adsorption of PO_4_^3−^-P reached equilibrium, but the increasing dosage had not broken the window of higher PO_4_^3−^-P removal efficiency.

PO_4_^3−^ + Al^3+^ = AlPO_4_ ↓
(4)


#### 3.1.2. Effect of PO_4_^3−^-P Concentration

The initial PO_4_^3−^-P concentration is a crucial factor for evaluating the adsorption performance of Al-ceramsite. Figure 2b shows the influence of initial PO_4_^3−^-P concentrations (2–10 mg/L) on PO_4_^3−^-P adsorption onto Al-ceramsite adsorbent. It was noticed that the PO_4_^3−^-P adsorption capacity rapidly increased to 256.18 mg/kg at initial PO_4_^3−^-P concentrations of 2–8 mg/L, and slowly increased to 274.68 mg/kg when the concentration of PO_4_^3−^-P increased to 10 mg/L. In addition, the removal efficiency of PO_4_^3−^-P for initial PO_4_^3−^-P concentrations increased from 75.42% to 82.61% with an increase in initial PO_4_^3−^-P concentrations from 2 to 10 mg/L. Al-ceramsite has high adsorption capacity for higher-concentration PO_4_^3−^-P solutions.

A large number of adsorption sites existed on the surface of Al-ceramsite. The PO_4_^3−^-P adsorption capacity was also increased with the increase in the solution PO_4_^3−^-P concentrations, indicating that PO_4_^3−^-P can migrate to the adsorption site of aluminum ceramsite faster, which shows better PO_4_^3−^-P removal efficiency [32].

#### 3.1.3. Effect of Temperature

Temperature played an essential part in the adsorption process and has a significant impact on the adsorption performance. From Figure 2c, the removal efficiency of PO_4_^3−^-P by Al-ceramsite showed an upward trend with temperature increased. The removal efficiency of PO_4_^3−^-P had achieved 81.27% within the temperature range of 25 °C. The adsorption capacity of Al-ceramsite on PO_4_^3−^-P increased from 274.71 to 295.12 mg/kg as the temperature increased from 25 °C to 35 °C, and then it did not exhibit a higher removal effect at 35 °C. Thus, it was revealed that PO_4_^3−^-P was quickly adsorbed on the Al-ceramsite at room temperature.

The adsorption capacity of PO_4_^3−^-P increasing with the temperature for Al-ceramsite demonstrated that PO_4_^3−^-P adsorption was an endothermic process. As discussed by Zhang et al., (2019), the temperature increase can accelerate the transfer rate of PO_4_^3−^-P to the adsorption site and can improve the contact probability between Al-ceramsite with PO_4_^3−^-P, which can improve the removal efficiency of PO_4_^3−^-P [33].

#### 3.1.4. Effect of pH

To determine the effect of the initial pH, the adsorption of PO_4_^3−^-P on the Al-ceramsite was achieved at pH = 5–9. It could be seen from Figure 2d that the initial pH had a certain impact on the removal effect of PO_4_^3−^-P. The higher PO_4_^3−^-P removal efficiency (92.81%) was achieved for Al-ceramsite at pH 5, and then decreased to about 66.45% as the initial pH increased from 5 to 9. The pH application window limited the removal efficiency of Al-ceramsite on PO_4_^3−^-P, which favored phosphate adsorption in the acidic environments.

The PO_4_^3−^-P removal efficiency would probably be determined by pH of the Al-ceramsite adsorption system, by the main form of PO_4_^3−^-P ionic species in the solution which is influenced by the different pH values (Equations (5) and (6)). In addition, the PO_4_^3−^-P is predominantly H_2_PO_4_^−^ and HPO_4_^2−^ in the solution. The positive-ions species could be facilitated the adsorption of the negative-ions species of PO_4_^3−^-P ions (H_2_PO_4_^−^ and HPO_4_^2−^) at the surface of Al-ceramsite, which further improved the removal efficiency of PO_4_^3−^-P by the Al-ceramsite [34]. The adsorption sites on the surface of Al-ceramsite could be limited during the competitive adsorption effect between OH^−^ and PO_4_^3−^ in the adsorption system; this means that the removal performance of PO_4_^3−^-P by Al-ceramsite was affected [35]. The positively charged ions on the surface of Al-ceramsite decreases with the increase of pH, thus weakening the electrostatic attraction and resulting in a decrease in the adsorption capacity of PO_4_^3−^-P [36].

H_3_PO_4_ ⇋ H_2_PO_4_^−^ + H^+^, pK_1_ = 2.12,
(5)


H_2_PO_4_^−^ ⇋ HPO_4_^2−^ + H^+^, pK_2_ = 7.20
(6)


### 3.2. Effect of Coexisting Ions on PO_4_^3−^-P Removal

Coexisting ions existed in the actual water environment, which may interfere with the efficiency of the adsorbent to remove PO_4_^3−^-P from the rainwater. Given that fact, in this study, the typical two cations (Ca^2+^, Mg^2+^) and four anions (SO_4_^2−^, CO_3_^2−^, F^−^, Cl^−^) were selected as coexisting ions to assess the effects of coexisting ions on PO_4_^3−^-P adsorption. Figure 3a illustrated that the adsorption capacity of Al-ceramsite increased in the presence of Ca^2+^ and Mg^2+^, which had a facilitating effect on the adsorption of PO_4_^3−^-P. However, SO_4_^2−^, CO_3_^2−^, F^−^, and Cl^−^ can affect the adsorption of PO_4_^3−^-P to compete with PO_4_^3−^ for the active adsorption site (Figure 3b). At the same concentration level, the existence of F^−^ had a stronger inhibitory effect on the adsorption of PO_4_^3−^-P than the other anions. When the initial concentration of F^−^ was 10 mmol/L, the adsorption capacity decreased 38.65% compared to the control, where the inhibitory effect of anion on the adsorption of PO_4_^3−^-P was achieved: F^−^ > CO_3_^2−^ > SO_4_^2−^ > Cl^−^.

Ca^2+^ and Mg^2+^ can improve the adsorption of PO_4_^3−^-P by Al-ceramsite, because they can be used as PO_4_^3−^adsorption sites and then form CaHPO_4_, Ca_3_(PO_4_)_2_, and Mg_3_(PO_4_)_2_ precipitates with PO_4_^3−^-P (Equations (7)–(9)), thus enhancing the retention of PO_4_^3−^-P by the Al-ceramsite porous surface [37]. SO_4_^2−^, CO_3_^2−^, F^−^, and Cl^−^ had an inhibitory effect on the adsorption of PO_4_^3−^-P, which could be due to the size of the hydrated ion radius affecting the adsorption of PO_4_^3−^-P [38]. In addition, the possibility exists that SO_4_^2−^ and Cl^−^ react with the reactive groups on the surface of Al-ceramsite to form outer-sphere complexes, and that CO_3_^2−^ and F^−^ could form inner-sphere complexes, thus producing competitive adsorption to inhibit PO_4_^3−^-P adsorption [39].

Ca^2+^ + HPO_4_^2−^ = CaHPO_4_↓,
(7)


3Ca^2+^ + 2PO_4_^3−^ = Ca_3_(PO_4_)_2_↓,
(8)


3Mg^2+^ + 2PO_4_^3−^ = Mg_3_(PO_4_)_2_↓
(9)


### 3.3. Desorption Characteristics of Al-Ceramsite

The prerequisites for the regeneration and reuse of Al-ceramsite depend on its adsorption–desorption potential. The PO_4_^3−^-P desorption effect of Al-ceramsite was evaluated at different pH conditions. Figure 4a illustrates the effect of different pH on the desorption of Al-ceramsite. The results revealed that the maximum release of PO_4_^3−^-P was 0.57 mg/L, at pH = 4 for 48 h. However, the release of PO_4_^3−^-P showed a linear increase at pH = 10, and it had reached 13.31 mg/L, which was 23.16 times higher than that at pH = 4. Under alkaline condition (pH = 10), the higher PO_4_^3−^-P desorption capacity of Al-ceramsite was achieved. Subsequently, the effects of desorbents (NaAC, Na_2_CO_3_, and NaOH) on the desorption of Al-ceramsite were investigated. It can be observed from Figure 4b that NaOH had the highest desorption efficiency for Al-ceramsite, which was much higher than that of Na_2_CO_3_ and NaAC. The effect of desorbents on PO_4_^3−^-P desorption were ranked as follow: NaOH > Na_2_CO_3_ > NaAC. Therefore, different concentrations of NaOH are used to evaluate the desorption efficiency of Al-ceramsite. The results are shown in Table 3. When the concentration of NaOH increased from 0.25 to 2 mol/L, the desorption rate of Al-ceramsite exhibits an increasing tendency. In general, when the concentration of NaOH was at 1 mol/L, the desorption capacity of Al-ceramsite could reach 70%. However, with NaOH concentration increased from 1 mol/L to 2 mol/L, the PO_4_^3−^-P desorption capacity of Al-ceramsite increased from 70% to 73%, and maintained stabilized. These results indicate that Al-ceramsite had an optimal desorption efficiency of PO_4_^3−^-P in a certain alkaline environment.

The pH value in the aqueous environment is a crucial factor not only for PO_4_^3−^-P adsorption, but also for PO_4_^3−^-P desorption. In an alkaline environment, the concentration of OH^−^ increased with increasing pH, and the increasing negative charge would weaken electrostatic attraction on the surface of Al-ceramsite to achieve PO_4_^3−^-P desorption [40]. Additionally, Al^3+^ and aluminum groups on the Al-ceramsite surface would be adsorbed by ligand exchange with PO_4_^3−^-P. The ligands and complexes would be further combined with the OH^−^ to form complexes or precipitates, releasing PO_4_^3−^, HPO_4_^2−^, and H_2_PO_4_^−^, thus achieving the desorption of Al-ceramsite again [41]. At a higher NaOH concentration, the desorption capacity of Al-ceramsite would not be further enhanced, which may be related to PO_4_^3−^-P adsorption forms, in which PO_4_^3−^-P adsorption was accompanied by irreversible chemisorption.

### 3.4. PO_4_^3−^-P Removal by Dynamic Adsorption of Al-Ceramsite

#### 3.4.1. Effect of Submerged Zone Depth on the Column

The dynamic adsorption column experiment based on Al-ceramsite is more reliable than the static adsorption experiment because it is similar to the stormwater biofilter. The feasibility of Al-ceramsite for the removal of PO_4_^3−^-P from stormwater was evaluated by column experiments. In the dynamic adsorption column, SZD can extend the hydraulic residence time (HRT), thus ensuring an excellent PO_4_^3−^-P removal performance. This study established 2, 5, and 8 cm SZDs for dynamic adsorption column experiments. In Table 4, when the run time increased from 24 h to 72 h, the PO_4_^3−^-P concentration gradually decreased at the same SZD. However, in the same run time, the effluent PO_4_^3−^-P concentration of different SZDs varied greatly, which could reach 53.49%. In column experiments with different SZDs, the effluent PO_4_^3−^-P concentration was lower than the environmental quality standards for surface water Class Ⅱ (0.1 mg/L), indicating that the adsorption effect of Al-ceramsite on PO_4_^3−^-P was more stable in different SZDs. When the SZD was applied to stormwater biofilters, the HRT could be extended, and Al-ceramsite could effectively cause the adsorption of PO_4_^3−^-P from the rainwater to achieve a higher PO_4_^3−^-P removal effect [42]. A too-high SZD will further affect the column effluent efficiency and may result in environmental problems. Therefore, the optimal SZD for this experiment was 5 cm.

#### 3.4.2. Effect of Filter Media Height on the Column

In the application of stormwater biofilters, the optimal FMH was determined because the FMH could affect the treatment effect and economic benefit. At the same as FMH, the effluent PO_4_^3−^-P concentration was relatively stable, as demonstrated in Table 5. However, with the increase of FMH, the dosage of Al-ceramsite further increased, whereas the effluent concentration of PO_4_^3−^-P gradually decreased. In addition, the effluent PO_4_^3−^-P concentration was lower than the environmental quality standards for surface water Class Ⅰ (0.02 mg/L) when the FMH exceeded 10 cm, and a further increase of FMH could not be achieved to improve the level of PO_4_^3−^-P concentration. Considering the effect of treatment effect and economic benefit, the optimal FMH was selected as 15 cm. In the column experiments, a larger adsorbent surface area was provided with the increased Al-ceramsite, and the removal efficiency of PO_4_^3−^-P was enhanced due to the formed aluminum phosphate by Al-ceramsite adsorption and immobilization of free aluminum and colloidal aluminum [43].

#### 3.4.3. Effect of Influent Flow in the Biofilter

The flow of rainwater into the stormwater biofilter could be change because of the randomness of rainfall magnitude under natural conditions. In order to further evaluate the effect of rainfall magnitude on PO_4_^3−^-P removal in the stormwater biofilter, the column experiments were simulated by controlled influent flow. Table 6 reflected that the dynamic adsorption column experiments could realize a higher removal efficiency of PO_4_^3−^-P from rainwater at the influent flow rate of 0.04–0.12 mL/s (24–72 h), and the effluent PO_4_^3−^-P concentration reached the environmental quality standards for surface water Class Ⅰ (0.02 mg/L).

The contact time between Al-ceramsite with PO_4_^3−^-P in rainwater biofilter could be affected by the influent flow. Shorter time and higher influent flow could reduce the membrane mass transfer resistance of the surface of Al-ceramsite. The mass transfer rate of PO_4_^3−^-P was accelerated, and the Al-ceramsite quickly reached a saturation state, thus realizing highly efficient removal of PO_4_^3−^-P [44]. The dynamic adsorption column experiments based on Al-ceramsite showed a higher and more stable PO_4_^3−^-P adsorption capacity, which could be applied in different rainfall conditions.

#### 3.4.4. Effect of Antecedent Dry Day on the Column

The treatment performance of the stormwater biofilter could be affected by ADD and influent flow during the actual natural rainfall. This study investigated the effect of PO_4_^3−^-P removal efficiency from rainwater at different ADD (3, 7, 14, 28 days) and different influent flow (0.04–0.12 mL/s). Figure 5 showed that the column experiment could achieve the complete removal of PO_4_^3−^-P from rainwater at 3–28 days of ADD, indicating that the column experiment based on Al-ceramsite had little effect on the removal of PO_4_^3−^-P at different ADD and that it had a wider application under different arid climatic conditions.

The impact of continuous drying events on contaminant removal could be prevented by the establishment of SZD in the column experiments [45]. Chandrasena et al. (2012) reported a decrease in contaminant removal performance of a sand-compost stormwater biofilter after an ADD, mainly attributed to fine fissures and macropore formation within the filler [46]. In the ADD, Al-ceramsite had little cracks and gaps compared with other composite fillers, further increasing the infiltration coefficient and reducing the contact of rainwater with the substrate, thus decreasing the treatment performance of the stormwater biofilter. In addition, it can be attributed to the high water-holding capacity of Al-ceramsite, which is due to the strong capillary force in the residual nano-pores of Al-ceramsite [47]. Also, the dry–wet cycle may stabilize and rinse the PO_4_^3−^-P in Al-ceramsite, which free the adsorption site of the Al-ceramsite and improve the PO_4_^3−^-P removal efficiency.

### 3.5. XPS of Ceramsite

The elemental composition and chemical state for the adsorbed PO_4_^3−^-P ceramsite and original ceramsite were investigated and revealed in the XPS spectroscopy in Figure 6. XPS of both Un-ceramsite and Al-ceramsite revealed the presence of C 1s, O 1s, Al 2p, and Fe 2p peaks, as presented in Figure 6a,b. Un-ceramsite showed a positive shift of 0.25 eV in the 3/2 orbital binding energy of Al 2p before and after adsorption (Figure 6c). In addition, the relative peak area of Al-ceramsite increased after adsorption (Figure 6d). It suggested that the PO_4_^3−^ interacting with Al^3+^ could occur by the ligand exchange during the adsorption process of Al-ceramsite [48]. The PO_4_^3−^ interacting with ceramsite was further studied by the change of the peak intensity of the phosphorus element after adsorption. Figure 6e,f observed that the binding energy of P 2p spectra of Un-ceramsite after adsorption is about 133.80 eV, while for Al-ceramsite after adsorption, the binding energy of about 133.05 eV of P 2p spectrum was found; compared with the standard P 2p spectrum of KH_2_PO_4_ (~134.0 eV), the binding energy of Un-ceramsite and Al-ceramsite P 2p spectrum was shifted ~0.20 eV and ~0.95 eV to a lower energy level, which confirmed the PO_4_^3−^-P adsorption by ceramsite. The change of binding energy further proved that the PO_4_^3−^-P interacting with aluminum formed a complex by the ligand exchange, thus enhancing the removal efficiency of PO_4_^3−^-P [49].

## 4. Conclusions

This study demonstrated the faster and higher PO_4_^3−^-P removal performance of Al-ceramsite compared to Un-ceramsite. Al-ceramsite could achieve up to 92.81% of PO_4_^3−^-P in static adsorption experiments (Al-ceramsite dosage of 40 g/L, initial PO_4_^3−^-P concentration of 10 mg/L, temperature of 25 °C, and pH of 5). The coexisting cations exhibited few negative effects on the adsorption of PO_4_^3−^-P by Al-ceramsite, while the presence of anions inhibited the adsorption of progress. The alkaline environment possessed a positive impact on the desorption of Al-ceramsite. According to the dynamic adsorption column experiments, PO_4_^3−^-P was completely removed from the actual rainwater while negligible effects were observed by influent flow and ADD. In addition, XPS of Al-ceramsite further described the adsorption of PO_4_^3−^-P. In conclusion, the development of Al-ceramsite-facilitated rainwater purification and solid waste resource recovery provides a favorable and sustainable way for PO_4_^3−^-P removal from stormwater. In future studies, the interactions of micro-organisms within the column on PO_4_^3−^-P removal were further verified to deepen the understanding of Al-ceramsite in practical applications.

## Figures and Tables

**Figure 1 ijerph-20-00671-f001:**
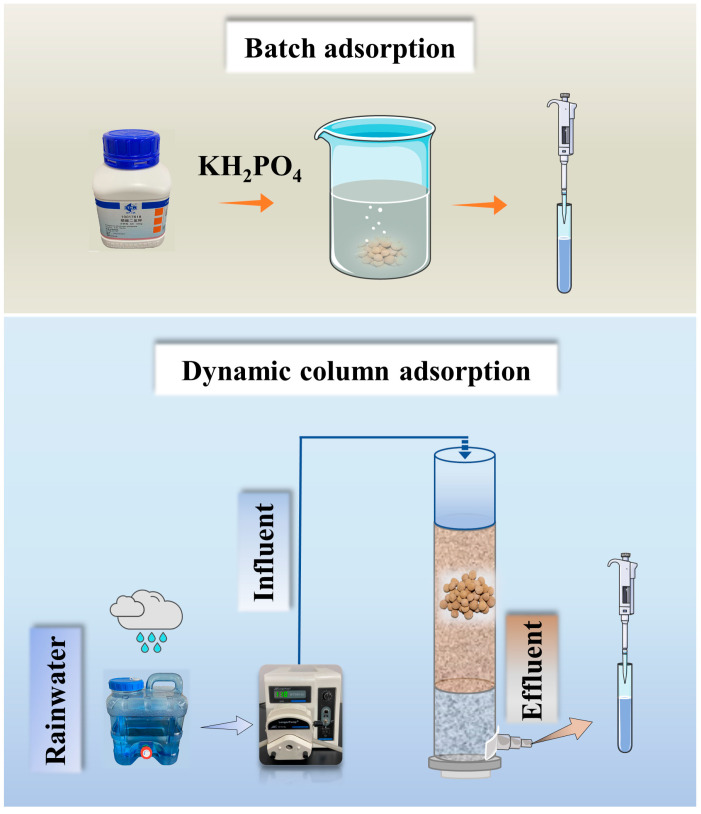
Schematic of the batch adsorption and column experiment.

**Figure 2 ijerph-20-00671-f002:**
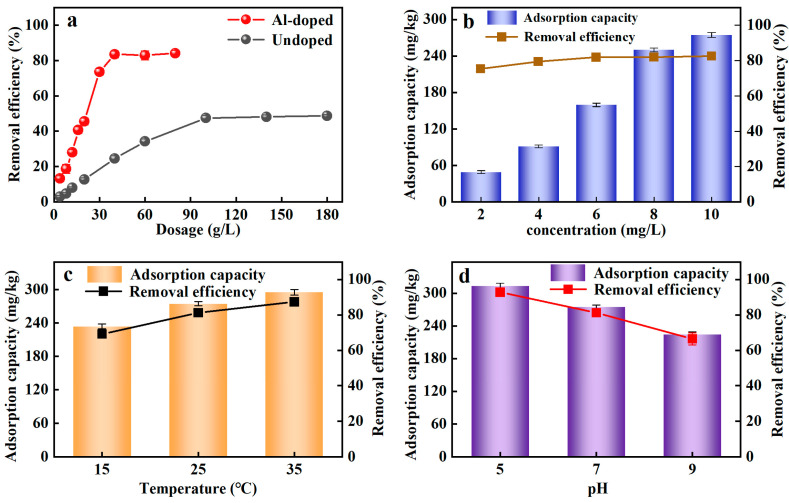
Adsorption efficiency of ceramsite under different conditions. (**a**): Al-doped and undoped; (**b**): initial PO_4_^3−^-P concentration; (**c**): temperature; (**d**): initial pH.

**Figure 3 ijerph-20-00671-f003:**
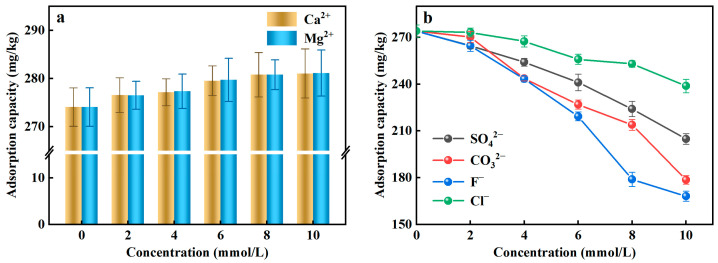
Effect of coexisting ions on adsorption of PO_4_^3−^-P by Al-doped ceramsite. (**a**): cationic; (**b**): anions.

**Figure 4 ijerph-20-00671-f004:**
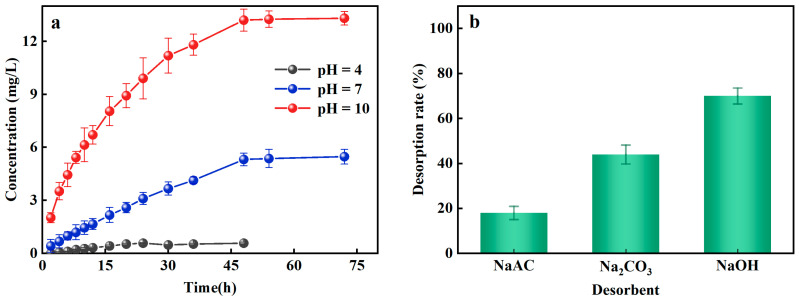
Effect factors on desorption of Al-doped ceramsite. (**a**): pH; (**b**): different desorbents.

**Figure 5 ijerph-20-00671-f005:**
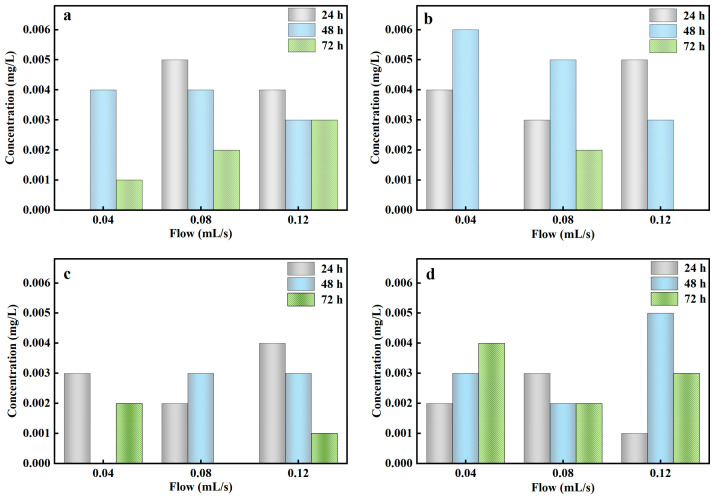
PO_4_^3−^-P concentration in different antecedent dry days. (**a**): 3 days; (**b**): 7 days; (**c**): 14 days; (**d**): 28 days.

**Figure 6 ijerph-20-00671-f006:**
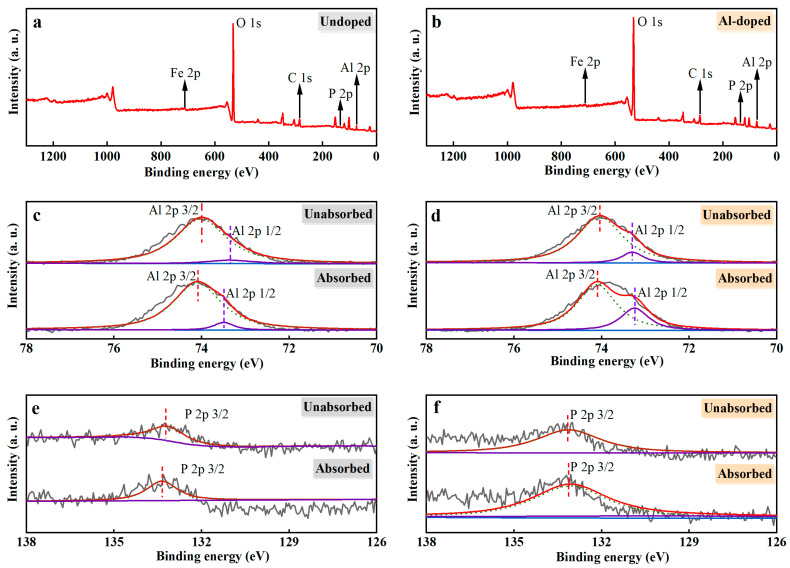
XPS spectra of ceramsite before and after adsorption. (**a**): widescan of undoped ceramsite; (**b**): widescan of Al-ceramsite; (**c**): Al elements spectra of Un-ceramsite; (**d**): Al elements spectra of Al-ceramsite; (**e**): phosphorus elements spectra of Un-ceramsite; (**f**): phosphorus elements spectra of Al-ceramsite.

**Table 1 ijerph-20-00671-t001:** Physicochemical properties of Un-ceramsite and Al-ceramsite.

Characterization	Un-Ceramsite	Al-Ceramsite
Particle size (mm)	4–6	4–6
BET surface area (m^2^/g)	1.71	5.31
Pore volume (cm^2^/g)	9.45 × 10^−3^	2.87 × 10^−2^
Average pore size (nm)	22.16	21.19
Al (wt%)	7.92	15.82

**Table 2 ijerph-20-00671-t002:** PO_4_^3−^-P adsorption by some typical adsorbents.

No. Adsorbent	Composition	Dosage (g/L)	Qe (mg/g)	Reference
Hangjin clay granular ceramic	Hangjin clay, montmorillonite, corn straw powders	10	5.96	[30]
Waste ceramsite	Coal fly ash, waterworks sludge, oyster shell	20	4.51	[27]
Modified bauxite residue	Gypsum, seawater, bauxite residue	40	0.35	[31]
La-ceramsite	Sewage sludge, coal fly ash, clay	20	0.104	[25]
Al-ceramsite	Sludge, cement block, coal fly ash	40	0.313	This study

**Table 3 ijerph-20-00671-t003:** Effect of NaOH concentration on desorption.

NaOH Concentration (mol/L)	Desorption Rate (%)
0.25	40
0.5	58
1	70
2	73

**Table 4 ijerph-20-00671-t004:** PO_4_^3−^-P concentration in different submerged zone depths.

Height (cm)	PO_4_^3−^-P Concentration (mg/L)				
	24 h	36 h	48 h	60 h	72 h
2	0.051	0.049	0.042	0.043	0.040
5	0.032	0.029	0.031	0.022	0.025
8	0.029	0.030	0.021	0.020	0.019

**Table 5 ijerph-20-00671-t005:** PO_4_^3−^-P concentration in different filter media heights.

Height (cm)	PO_4_^3−^-P Concentration (mg/L)				
	24 h	36 h	48 h	60 h	72 h
5	0.032	0.029	0.031	0.022	0.025
10	0.017	0.013	0.018	0.011	0.010
15	0.005	0.009	0.001	0	0
20	0.004	0.006	0.003	0	0

**Table 6 ijerph-20-00671-t006:** PO_4_^3−^-P concentration in different influent flow.

Flow (mL/s)	PO_4_^3−^-P Concentration (mg/L)		
	24 h	48 h	72 h
0.04	0.003	0.001	0.002
0.08	0.004	0	0.001
0.12	0.002	0.004	0.002

## Data Availability

The data are included in the article and available upon request.

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
