# Peer review of "Adsorption Characteristics of Phosphate Based on Al-Doped Waste Ceramsite: Batch and Column Experiments"

_ijerph, 2022, doi:10.3390/ijerph20010671_

Round 1

Reviewer 1 Report

This work examined the use of the Al-doped waste ceramsite for the adsorption of phosphate. The manuscript is interesting and can be considered for publication in International Journal of Environmental Research and Public Health with a major revision. My specific comments are as follows:

1.       The authors should carefully check the language use of the entire submission.

2.       Line 63-64: Please explain the source of the major elements in sludge/coal fly ash/cement blocks.

3.       Line 96: I suggested to add the previous study literature to reflect the preparation of Al-ceramsite of this experiment.

4.       Line 116: Please explain whether to use rainwater after collection or immediately?

5.       Line 122: Please explain why Al-ceramsite and other materials are present in the column experiment.

6.       Line 193: It is suggested to add the effect of PO43--P concentration on the adsorption of Al-doped waste ceramsite.

7.       Line 201: " With the increased initial PO43--P concentrations..." This sentence could be rewritten better.

8.       Line 204: I don't see the removal efficiency of PO43--P in Figure 2b, please add clarification.

9.       Line 223: " which favored phosphate adsorption in the acidic environments..." Please explain and add to the literature discussion.

10.   Line 237: Please explain why selected Ca2+, Mg2+, SO42-, CO32-, F-, Cl- as coexisting ions.

11.   Line 264: It is not clear whether the Al-ceramsite can be reused, so it is suggested to supplement the repeatability experiment.

12.   Line 331-333: Please explain why the authors choose the influent flow rate of 0.04-0.12 mL/s as the rainfall level.

13.   Line 346: In my opinion, the antecedent dry day can affect the column experiment. Please explain and add to the literature discussion.

14.   It is suggested to add the content of comparing the data of column dynamic adsorption with other literature to reflect the efficiency of this experiment.

15.   Details should be improved: some languages in the whole manuscript need to be revised, and the format of references should be unified.

Reviewer 2 Report

Phosphorus are widely used in modern agriculture. Monitoring of this type of pollution should be carried out in waters and rivers. The simple  methods for removal phosphate are currently being sought. 

I have reviewed the  manuscript entitled: .

Adsorption characteristics of phosphate based on Al-doped  waste ceramsite: Batch and column experiments”.

In my opinion the manuscript need minor revision.

Comment1

Line 100 How long the samples were calcined at 1150 oC?.

Comment 2

Line 116 What other ions besides phosphate were in the rainfall?

Comment 3

Line 146 What was the linear flow rate in the column?

Comment 4

Line 152 What was the limit of phosphate detection?

 Comment 5

Line 336 There is ml/s it should be mL/s.  Please standardize the notation of units.

Comment 6

Line 322 Was the concentration of aluminum ions determined in the effluent?

Reviewer 3 Report

The importance and novelty of this study should be more highlighted in the introduction section.
